# Acute Middle Cerebral Artery Occlusion Detection Using Mobile Non-Imaging Brain Perfusion Ultrasound—First Case

**DOI:** 10.3390/jcm11123384

**Published:** 2022-06-13

**Authors:** Mustafa Kilic, Christina Wendl, Sibylle Wilfling, David Olmes, Ralf Andreas Linker, Felix Schlachetzki

**Affiliations:** 1Department of Neurology, Center for Vascular Neurology and Intensive Care, University of Regensburg, University Hospital Regensburg, Medbo Bezirksklinikum Regensburg, Universitaetsstr. 84, 93053 Regensburg, Germany; mustafa.kilic@medbo.de (M.K.); sibylle.wilfling@medbo.de (S.W.); david.olmes@medbo.de (D.O.); ralf.linker@medbo.de (R.A.L.); 2Center for Neuroradiology, University Hospital Regensburg, Medbo Bezirksklinikum Regensburg, Universitaetsstr. 84, 93053 Regensburg, Germany; christina.wendl@ukr.de

**Keywords:** prehospital stroke diagnosis, large vessel occlusion, ultrasound, thrombectomy, brain perfusion, SONAS^®^, prehospital stroke scales, point-of-care diagnostics

## Abstract

Mobile brain perfusion ultrasound (BPU) is a novel non-imaging technique creating only hemispheric perfusion curves following ultrasound contrast injection and has been specifically designed for early prehospital large vessel occlusion (LVO) stroke identification. We report on the first patient investigated with the SONAS^®^ system, a portable point-of-care ultrasound system for BPU. This patient was admitted into our stroke unit about 12 h following onset of a fluctuating motor aphasia, dysarthria and facial weakness resulting in an NIHSS of 3 to 8. Occlusion of the left middle cerebral artery occlusion was diagnosed by computed tomography angiography. BPU was performed in conjunction with injection of echo-contrast agent to generate hemispheric perfusion curves and in parallel, conventional color-coded sonography (TCCS) assessing MCAO. Both assessments confirmed the results of angiography. Emergency mechanical thrombectomy (MT) achieved complete recanalization (TICI 3) and post-interventional NIHSS of 2 the next day. Telephone follow-up after 2 years found the patient fully active in professional life. Point-of-care BPU is a non-invasive technique especially suitable for prehospital stroke diagnosis for LVO. BPU in conjunction with prehospital stroke scales may enable goal-directed stroke patient placement, i.e., directly to comprehensive stroke centers aiming for MT. Further results of the ongoing phase II study are needed to confirm this finding.

## 1. Introduction

Mechanical thrombectomy (MT) has evolved as the most efficient therapy in cerebral ischemic stroke due to large vessel occlusion (LVO) achieving a 20–27% absolute increase in patients regaining functional independence compared to patients without receiving MT [1,2]. These impressive results, however, are being achieved by rigorous patient selection by cerebral perfusion imaging, achievement of successful recanalization and short symptom to recanalization times including transport to comprehensive stroke centers (CSC) [3,4]. In rural areas, stroke units (SU) and telestroke networks have been established only with intravenous thrombolysis (IVT) capabilities more than 20 years ago, and further transfer to CSC for MT is time-consuming, causing delay in the initiation of recanalization therapy.

Still, prehospital stroke detection algorithms employ clinical scales deemed imperfect for detecting LVO. Thus, transfer of severe stroke patient to either CSCs or the next regional SU remains a complex decision [5]. Ultrasound is currently the only mobile imaging modality widely available for brain vascular diagnostics. Our group demonstrated that transcranial color-coded sonography (TCCS) using mobile color-Duplex point-of care ultrasound (POCUS) can be performed in the prehospital phase with very high sensitivity and specificity for the detection of middle cerebral artery (MCA) occlusion [6,7]. Another highly efficient alternative, the mobile stroke units (mSU) with computed tomography (CT), including CT-angiography installed in an ambulance to perform at the patient’s site, is of limited availability, costly and has a rather small range. The air-mobile SU version of the mSU is still in the concept phase [8,9]. However, the drawback of TCCS is the high neurological expertise requiring extensive practical and theoretical training programs for paramedics to employ the pathophysiological driven diagnostic concept for TCCS, i.e., focusing on the right middle cerebral artery in left-sided hemiparesis and neglect [10]. 

We recently published the results of a phase I study using a mobile battery driven brain perfusion ultrasound (BPU) device SONAS^®^ (BURL Concepts, Inc., San Diego, CA, USA) in healthy volunteers with good correlation of measured hemispheric brain perfusion to perfusion weighted MRI [11]. This case report describes the BPU finding in a patient with acute MCA occlusion confirmed by CT-angiography and TCCS prior to MT. 

## 2. Materials and Methods

During initiation of a phase I–II study at the University of Regensburg, Department of Neurology at the medbo Bezirksklinikum (approval by the local ethics committee, IRB protocol number 2018-001279-19, in accordance with the World Medical Association Declaration of Helsinki), an acute stroke patient was admitted and individual informed consent in the presence of his parents as well as post-hoc consent for publication 1 year later given. The phase I–II study SONAS enrolled acute stroke patients with perfusion CT and was registered at ClinicalTrials.gov (NCT03897153). In brief, patients with acute ischemic stroke within 24 h will be investigated using the SONAS^®^ device after confirmation of an either proximal, middle or distal MCA main stem occlusion or distal internal carotid artery occlusion (including carotid-T occlusion). Confirmation of the LVO will be performed by cerebral magnetic resonance imaging (cMRI) or cerebral imaging computed tomography (cCT), including perfusion weighted (pw) imaging sequences. The study has now been completed and data analysis is in preparation. 

### 2.1. Case Report

The 46-year-old male patient was transferred for suspected left hemispheric ischemia as on admission, he presented with moderate dysarthria, motor aphasia and mild facial paresis (NIHSS 3) starting early in the morning. He reported that about 4–5 weeks prior, one side of the face had felt strange for 10–15 min, suggestive of a transient ischemic attack. Admission was approximately 12 h after stroke onset. At this time, cCT was almost unremarkable (ASPECTS Score 9), while CT-angiography revealed proximal MCA occlusion (M1 segment) on the left side with good collateralization (collateral score 3–4, Figure 1). With fluctuating symptoms between progressive global aphasia and brachiocephalic hemiparesis (NIHSS 3 to 8), and an overall deficit relevant to daily living, the decision was made for MT, especially since no intravenous thrombolysis was initiated. Informed consent for additional ultrasonography was obtained from the patient and his parents and retrospectively. Both TCCS and BPU were performed in parallel to MT preparation in general anesthesia without causing a time delay. 

Digital subtraction angiography demonstrated proximal MCA occlusion with good cortical collaterals via leptomeningeal anastomosis predominantly from the anterior cerebral artery (Figure 2). After successful MT (TICI 3), the patient was extubated and presented only mild word finding difficulties and minimal facial paresis (NIHSS 2). On telephone follow up two years later, he was back to work and neurologically unremarkable (modified Rankin Scale 0).

### 2.2. Transcranial Color-Coded Sonography (TCCS)

TCCS was performed using a standard color Duplex ultrasound system equipped with a low-frequency phased array transducer (Philips, Affiniti 70, S5-1 cardiac sector probe, Amsterdam, The Netherlands). Transtemporal insonation for detection of the basal cerebral arteries was performed as previously described [12]. In brief, after detection of a sufficient temporal bone window by identification of the mesencephalic peduncle and the contralateral skull in B-mode color Duplex mode helped to identify patent cerebral arteries with flow towards the transducer depicted in red, and flow away from the transducer in blue. Pulsed Doppler measurements were performed to quantify flow velocities. An occluded MCA can be assumed when the anterior cerebral artery can be depicted confirming a diagnostic useful temporal bone window.

### 2.3. Brain Perfusion Ultrasound (BPU) Using SONAS^®^

BPU was performed as previously described employing the CE-certified (Class IIa), non-imaging ultrasound device SONAS^®^, a portable, battery-powered device to generate hemispheric time intensity curves after injection of an ultrasound enhancing agent, which in this case was SonoVue^®^ [11,13]. In brief, bilateral low-frequency transducer with low transmit frequencies and power (220 kHz, 2% alternating duty cycle from right to left, TIC and MI < 1.0) measure the 4th to 6th harmonic frequencies generated by intravenous 2.4 mL microbubble injection. Time to peak (TTP) curves are generated based on measurements contralateral, that is transmission and reception are on opposite sides of the head, and ipsilateral measurement, where transmission and receiving are ipsilateral (Figure 3). Using an automatic peak detection algorithm for both brain hemispheres separately, hemispheric TTP values are compared. The differential of peaks is computed and shown as a delta-TTP (dTTP) value. 

## 3. Results

Prior to embolectomy and 30 min after neurological deterioration, both, BPU und TCCS were performed. BPU demonstrated significant delay in left hemispheric perfusion with a delay of 6.6 s (Figure 4).

Conventional TCCS demonstrated only only slow proximal pseudo-venous flow in the proximal MCA (peak systolic flow less than 10 cm/s) consistent with MCA occlusion and relatively normal flow in the left ACA (Figure 5).

### Case Report Continued

Decision for embolectomy under general anesthesia was made based on the persistent MCA occlusion and fluctuating course of neurological deficits (NIHSS 3 to 8). The first angiographic image was made at 20:45 h and complete TICI 3 recanalization was noted at 21:17 h. After extubation, NIHSS score was 3 and 2 in the following days. Neuropsychological testing showed mild cognitive impairment and indication for neurorehabilitation. At telephone follow-up 2 years later, the patient was again able to work and independent in daily life (mRS score of 0–1).

## 4. Discussion

This is the first report describing the application of BPU in a patient with acute MCA occlusion shown by CT-angiography and later digital subtraction angiography using a portable POCUS device for BPU (SONAS^®^) with comparison to standard TCCS as another mobile method for acute stroke diagnosis, while TCCS allowed MCA occlusion imaging with similar information as CT-angiography and DSA BPU depicts the subsequent brain perfusion changes. If replicated in the ongoing phase II study, BPU may be feasible in prehospital stroke diagnostics. 

The progressive developments in the field of MT with high recanalization rates along with decreasing peri-interventional risk challenge the identification and allocation of acute stroke patients in the prehospital phase. In stroke patients in need for MT, direct transfer to an endovascular MT-capable center may be preferred as opposed to transport to the closest primary stroke center and secondary transport for MT [14]. To date, efforts have been made to identify LVO employing prehospital stroke scales as shown in the PRESTO study comparing eight scales employed by paramedics in the prehospital phase and identifying three scales with acceptable-to-good accuracy [15]. In another prospective study on patients with ischemic stroke entered in the Dyon stroke registry, Duloquin et al. applied 16 different prehospital stroke scales; however, a priori excluding patients with ICH severely reducing its value as a prehospital study [16]. Even with this drawback sensitivities of scales were still low (identification of LVO ranging between 0.64 to 0.79, sensitivity 59% to 93%, and specificity ranging from 34% to 89%, c-statistic) with 174 of 971 patients (17.9%) had LVO (defined as occlusion of M1 and M2-segment of the MCA and basilar artery occlusion). In a topical review from 2020, van Gaal and Demchuck listed common flaws of studies on the validity of prehospital stroke scales, among them, exclusion of ICH patients, lack of prospective studies in the field as opposed to application of stroke scales to pure ischemic stroke databases (exclusion, i.e., TIA, ICH, stroke mimics, amongst others), and whether or not these scales have indeed been tested by paramedics [17]. However, in a recent pilot study in the Baltimore metro area patients were re-routed CSCs for MT upon application of the Los Angeles Motor Scale. Consequently, significantly shorter procedural times from symptom to MT onset of 119 min were achieved [18]. While the faster initiation of MT showed a strong non-significant trend for better outcome, more than 50% of patients were allocated to the wrong clinic. 

However, fast and goal-directed prehospital point-of-care diagnostics such as blood serum biomarkers and TCCS have been advocated to further accelerate the diagnostic and therapeutic stroke pathways in stroke patients [19,20,21,22]. Apart from mobile stroke units, only portable TCCS is a neuroimaging tool capable of visualizing LVO in the field, preferentially employing echo-enhancing agents [23,24]. In the patient described here, conventional TCCS identified left MCA occlusion in line with CT-angiography and digital subtraction angiography. BPU, which involves single-slice transcranial ultrasound using phased-array transducers and echo-enhancing agents to produce perfusion maps comparable to perfusion CT, is not currently mature enough to be used for routine practice, let alone prehospital use [25,26]. Yet, for conventional TCCS, detailed pathophysiological hypothesis of the stroke location and hands-on expertise in identifying an adequate temporal bone window is needed. However, many of these obstacles may be solved by employing telemetric and artificial intelligence support [27,28]. BPU using the SONAS^®^ device differs significantly from TCCS and other transcranial ultrasound techniques as it lacks anatomical information and presents only perfusion curves [11]. However, the demonstrated delay in time-to-peak times of the left hemisphere together with the clinical information can be easily gathered with little expertise, ideal for application in the prehospital phase. Table 1 compares the basic findings, techniques, advantages and disadvantages of TCCS and non-imaging BPU using SONAS^®^.

## 5. Conclusions

This is the first patient with MCA occlusion in whom both the BPU technique with the SONAS^®^ device and standard TCCS were used to visualize MCA occlusion and the resulting perfusion deficit. The results of BPU using the SONAS^®^ device were compared with conventional imaging (CT angiography, TCCS and DSA) and revealed a similar result. The user-friendly mode of operation of BPU requiring little additional skills indicate the potential of SONAS^®^ to fasten stroke treatment when positively identifying LVO at the earliest time point possible after symptom onset. Goal-directed hospital admission for MT and fewer secondary transportations may enable faster symptom to recanalization times, and eventually better outcome. The results of the ongoing phase II study with the SONAS^®^ device in acute ischemic stroke patients as well as further studies on lacunar infarction, intracerebral hemorrhage, minor strokes and stroke mimics are the prerequisite to define the full potential of BPU.

## Figures and Tables

**Figure 1 jcm-11-03384-f001:**
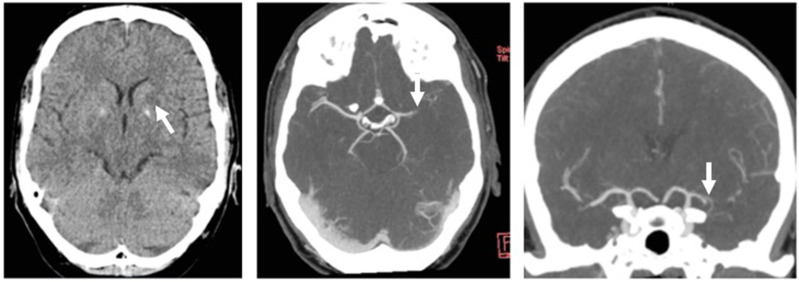
**Left**: unenhanced CT, putaminal hypodensity (arrow), ASPECTS score 9; **middle**: axial maximum intensity projection (MIP) reconstruction of CT-angiography showing left MCA occlusion (arrow); **right**: coronal MIP reconstruction of left MCA occlusion (arrow), good leptomeningeal collaterals can be seen.

**Figure 2 jcm-11-03384-f002:**
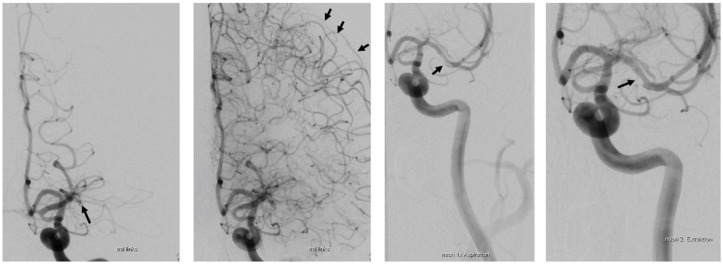
Digital subtraction angiography. **Left** to **right**: phase after contrast injection showing proximal MCA occlusion (arrow); demonstration of extensive leptomeningeal collaterals arising from the ACA (arrows); immediately after mechanical thrombectomy by first use of the stent retriever showing suspicious residual thrombus with mild lumen narrowing (arrow) and, right, complete recanalization with mild stenosis/vasospasm (TICI 3).

**Figure 3 jcm-11-03384-f003:**
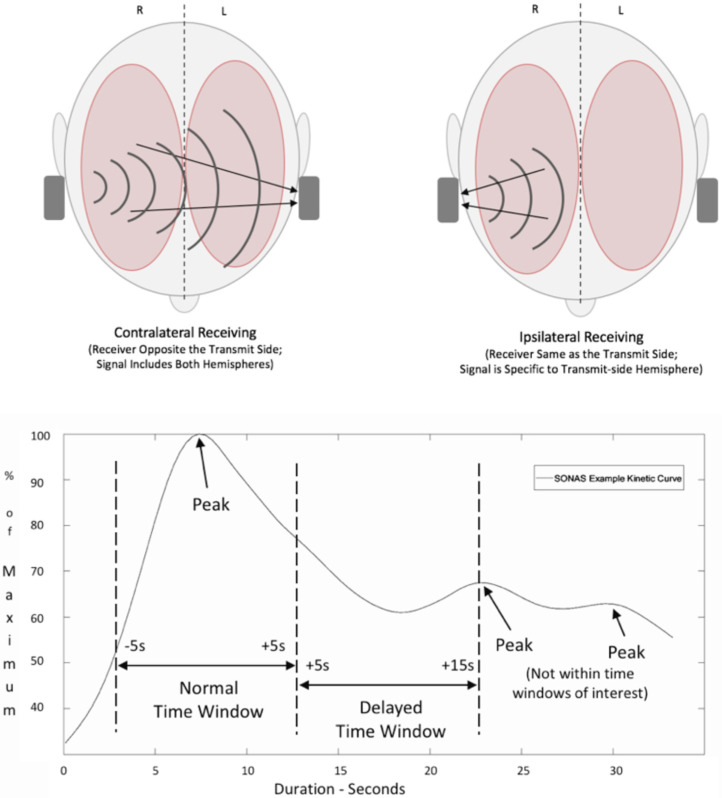
Upper figure: signal transmission and recording from both ultrasound transducers after injection of echo-enhancing agents demonstrating the mode of operation. R—right, L—left. Lower figure: time intensity curves after contrast agent injection with X-axis time after injection and Y-axis maximum signal backscatter analyzed automatically by BPU machine (SONAS^®^). The first peak is due to the first appearance of contrast agent while the second and third peaks are to due systemic reflow phenomenon.

**Figure 4 jcm-11-03384-f004:**
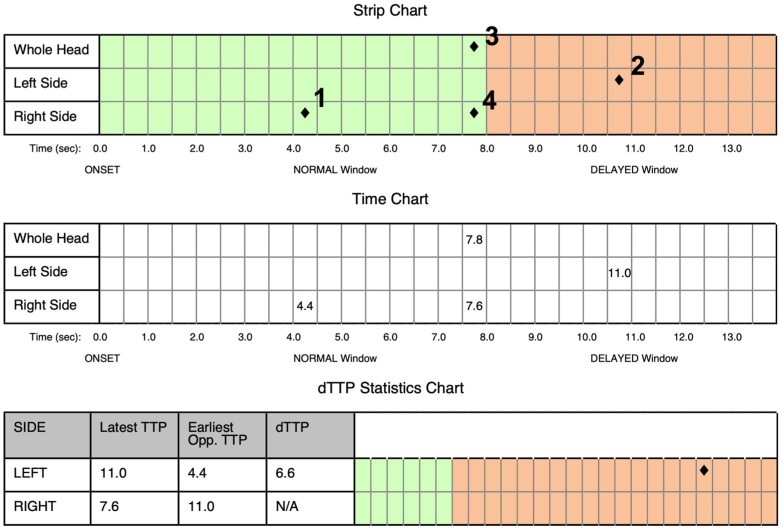
Analysis of time intensity curves representing brain perfusion ultrasound using SONAS^®^. Upper row: 1—first time-to-peak (TTP) in the “healthy” right hemisphere, 2—delayed left hemispheric peak, 3—peak of the whole brain, and 4—delayed second peak probably representing right anterior cerebral artery crossflow as the primary collateral pathway. Middle row: absolute time values of the peaks. Lower row: expression of the difference in time-to-peak values (delta-TTP) expressing the absolute perfusion deficit in the left hemisphere.

**Figure 5 jcm-11-03384-f005:**
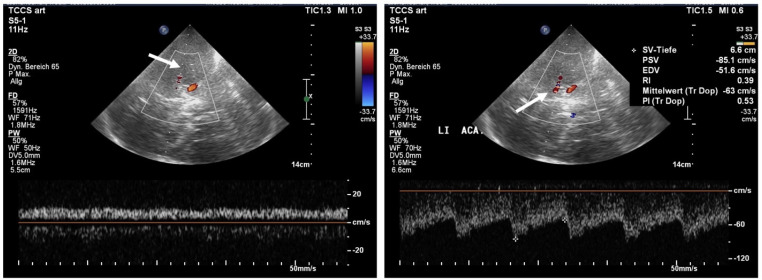
TCCS scan through the left temporal bone window with transcranial color-coded sonography in the upper and Doppler spectrum from the Doppler gate. **Left**: Doppler gate placed at the origin of the middle cerebral artery with absent color filling (arrow) in the M1 segment and pseudo-venous flow at the beginning of the M1-Segment (Doppler gate depth 55 mm). **Right**: anterior cerebral artery color-coded in red due to aliasing due to increased flow velocities with the Doppler gate placed at a depth of 66 mm in the A1-segment showing increased flow of 63 cm/s mean maximum flow and decreased resistance index of 0.39 all suggestive of collateral flow.

**Table 1 jcm-11-03384-t001:** Comparison of TCCS and BPU.

	Brain Perfusion Ultrasound	Transcranial Color-Coded Sonography
Information	Hemispheric brain perfusion	Ipsilateral intracranial arteries
Operator qualification	Low due to automated analysis	High, requires anatomical knowledge for interpretation
Controls	Single button, sequential operation	Variable keyboard complexity, pro–grammable presets
Bone penetration	High	Low to high, can be increased using echo-enhancing agents
Echo-enhancing agents	Required	Optional
Presentation	Generation of time-intensity curves on-site	Real-time depiction of vessel occlusion
Potential additional information	Raised intracranial pressure	Intracerebral hemorrhage Brain parenchymal shift (i.e., midline displacement, intracerebral hemorrhage, hydrocephalus)

## Data Availability

The raw data underlying this article are intended for publication on a suitable platform and can be made available by the corresponding author on reasonable request.

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
