# Peer review of "Acute Middle Cerebral Artery Occlusion Detection Using Mobile Non-Imaging Brain Perfusion Ultrasound—First Case"

_jcm, 2022, doi:10.3390/jcm11123384_

Round 1

Reviewer 1 Report

xx 

The authors report the impact of mobile non-imaging brain perfusion ultrasound on detecting acute middle cerebral artery occlusion.

“Continuous oxygen delivery and CO2 clearance are paramount in the maintenance of normal brain function and tissue integrity[1]. In patient with stroke,  the impairment of cerebral autoregulation and decreased blood flow results in delayed ischemic injury[2]. timely detection of neurological deterioration[3], perfusion deficits are important issue. The invention of the X-ray is an important event in the very early 20th century[4]. High-level techniques have been used in current medical practice[5]. This is a case report, such a study is important because it identifies the mechanism and provides impetus to future research[6]. Only a few studies have been reported on mobile non-imaging brain perfusion ultrasound. The manuscript (without references)  is similar to another article at a rate of 39%. It should be less than 30%.

12% of the manuscript was taken from only one article   (https://doi.org/10.1186/s42466-022-00179-8)It should be less than 5%.

References

1.          Kanat A (2003) Brain oxygenation and energy metabolism: Part 1--Biological function and pathophysiology. Neurosurgery 52:1508–9; author reply 1509

2.          Ozdemir NG, Aydin MD, Yolas C, et al (2017) Predictive role of external carotid artery vasospasm on cerebral ischemia after subarachnoid hemorrhage: Experimental study. Turk Neurosurg 27:874–883. https://doi.org/10.5137/1019-5149.JTN.17206-16.2

Author Response

The authors like to thank reviewer 1 for the positive comment on the importance of a novel technique assessing brain perfusion by means of novel non-imaging point-of-care device. The only publication on non-imaging brain perfusion ultrasound using the SONAS® device is reference 11, in fact a publication from our group with the same first and last author. Thus, wording habits may lead to some similarity and we did our best to avoid repetition in the revised manuscript. All changes can be identified in the version with track-changes marked.

Reviewer 2 Report

The authors present a case report describing their use of a brain perfusion ultrasound to confirm a MCA occlusion in a patient. Although the manuscript is well written, there are some items that need to be addressed prior to publication.

There should be more explanation for the rationale behind intervening on a patient with a low NIHSS. Was the patient young? They stated that the symptoms were fluctuating but were they medically managed and still variable? When they worsened to NIHSS 8, was the decision made to take them for MT then? If so, why describe them as a NIHSS 3?

Figure 3 should be better labeled to understand what is being shown.

Figure 4 needs to be explained better as it is not clear to understand the chart from the figure legend alone. In fact, this chart is the main novelty of the manuscript and should likely have a part of the discussion dedicated to interpretation of the results of the BPU.

Page 6 -line 169 Unclear why the authors are using the word “extruding” here? The sentence itself doesn’t make sense. Please clarify.

Overall I believe this paper is of value but these concerns need to be addressed prior to acceptance.

Author Response

The authors present a case report describing their use of a brain perfusion ultrasound to confirm a MCA occlusion in a patient. Although the manuscript is well written, there are some items that need to be addressed prior to publication.

Answer: The authors like to thank the reviewer for this positive comment and the opportunity to re-submit a revised version.

There should be more explanation for the rationale behind intervening on a patient with a low NIHSS. Was the patient young? They stated that the symptoms were fluctuating but were they medically managed and still variable? When they worsened to NIHSS 8, was the decision made to take them for MT then? If so, why describe them as a NIHSS 3?

Answer: The patient was indeed fairly young – age 46 – and while he was admitted with NIHSS 3 he did fluctuate to NIHSS 8 with a speech dominant stroke. Since no IV thrombolysis could be initiated due to significant delay in presentation we did not risk waiting for an unlikely spontaneous recanalization as we confirmed persistant MCA occlusion by means of TCCS. We re-formulated the clinical rational for our decision making and added this imformation.

Figure 3 should be better labeled to understand what is being shown.

Answer: This is a standard TCCS image and we added more information as requested.

Figure 4 needs to be explained better as it is not clear to understand the chart from the figure legend alone. In fact, this chart is the main novelty of the manuscript and should likely have a part of the discussion dedicated to interpretation of the results of the BPU.

Answer: Thank you for this important remark as non-imaging brain perfusion ultrasound is a novel technique. We did our best and added elaborated on this in the figure legend plus revised the image.

Page 6 -line 169 Unclear why the authors are using the word “extruding” here? The sentence itself doesn’t make sense. Please clarify.

Answer: Thank you so much for finding this mistake! Extruding was replaced to “identifying” and now the sentence does make sense.

Reviewer 3 Report

A well written case report describing acute middle cerebral artery occlusion and detection with mobile brain perfusion ultrasound. The interesting case is well presented and the results, discussion and conclusions are sound. It presents a promising introduction to the phase II study about the usefulness of brain perfusion ultrasound in detecting and treating cerebral artery occlusion.

Author Response

Thank you very much for this positive evaluation.